# The Neglected Angio-Neurotrophic Parasite *Gurltia paralysans* (Nematoda: Angiostrongylidae): Northernmost South American Distribution, Current Knowledge, and Future Perspectives

**DOI:** 10.3390/pathogens10121601

**Published:** 2021-12-09

**Authors:** Manuel Uribe, Sara López-Osorio, Jenny J. Chaparro-Gutiérrez

**Affiliations:** 1CIBAV Research Group, Veterinary Medicine School, Universidad de Antioquia, Medellín 050034, Colombia; mmanuel.uribe@udea.edu.co (M.U.); sara.lopezo@udea.edu.co (S.L.-O.); 2Biomedical Research Center Seltersberg (BFS), Institute of Parasitology, Justus Liebig University Giessen, 35392 Gießen, Germany

**Keywords:** *Gurltia paralysans*, gurltiosis, paralysis worm, metastrongyloid, feline, South America

## Abstract

*Gurltia paralysans* is a rare metastrongyloid nematode in South America that has begun to gain relevance in feline internal medicine as a differential diagnosis of progressive degenerative myelopathy disorders. The parasite life cycle has not been fully elucidated but probably involves invertebrate gastropod fauna as obligate intermediate hosts; thus, *G*. *paralysans* remaining an extremely neglected parasitosis. Feline gurltiosis *intra vitam* diagnosis is highly challenging due to lack of evidence in the excretion of *G. paralysans* eggs and larvae, neither in feces nor in other body secretions because environmental stages and the transmission route of the parasite remain unknown. Unfortunately, no experimental trials for the treatment of feline gurltiosis have been conducted to date. However, there are some reports of the successfully antiparasitic drugs used with different effectiveness and clinical improvement results in diagnosed cats. Further studies are needed to evaluate the parasite occurrence among domestic cats and the neotropical wild felid species distributed within Colombia in addition to the gastropod fauna that may harbor the developing larvae (L1–L3) stages of this underestimated parasite.

## 1. Introduction, Brief History, and the Enigmatic Life Cycle

Tropical and subtropical geographic areas offer appropriate ecological characteristics for re-emergence, maintenance, and dissemination of multiple parasitic infectious diseases, with some of them extremely neglected [1,2,3]. Rare and underestimated felid parasites such as the nematodes *Dracunculus insignis* (Dracunculidae) [4], *Brugia malayi* (Onchocercidae) [5,6], *Thelazia callipaeda* (Thelaziidae) [7,8], *Ollanulus tricuspis* (Molineidae) [9,10], *Physaloptera praeputialis* (Physalopteridae) [11], *Oslerus rostratus* (Filariodidae), *Troglostrongylus subcrenatus* (Crenosomatidae), and *Angiostrongylus chabaudi* (Angiostrongylidae) [12]; the trematode *Amphimerus sp. (Opisthorchiidae)* [13]; and the cestode *Mesocestoides corti* (Mesocestoididae) [14] have been documented in domestic cats (*Felis catus*) in the Americas, Asia, and Europe. Additionally, *Gurltia paralysans* is a rare metastrongyloid nematode in South America that has begun to gain relevance in feline internal medicine as a differential diagnosis of progressive degenerative myelopathy disorders. However, it is still neglected and underestimated [15]. This angio-neurotrophic nematode is infrequently described, often unnoticed, and out of the diagnostic scope of even specialist veterinary clinicians, similarly to other neglected parasitoses [16,17]. The poorly understood felid angio-neurotrophic parasite was first classified as *Hemostrongylus*, then as *Angiostrongylus*, and finally located as the only species of the genus *Gurltia* (Nematoda: Angiostrongylidae). The first description of *G. paralysans* took place in the temperate forest ecoregion of Valdivia in southern Chile by Wolffhügel in 1933 [18]. The nematode genus was named in honor of both Dr. Ernst Friedrich Gurlt (1794–1882) and the species *paralysans*, due to observed paralysis being the main clinical sign in infected felids [16,18]. This metastrongyloid nematode was reported parasitizing domestic cats (*F. catus*) in southern Chile, and the small wild kodkod (*Leopardus guigna*) has been proposed as the natural host [16,19]. The parasite has been reported in other wild felids such as the margay (*Leopardus wiedii*) [20]. Moreover, the Goeffroy’s cat (*Leopardus geoffroyi*) and the northern tiger cat (*Leopardus triginus*) have also been proposed as potential native final hosts [15,21].

The nematode life cycle has not been fully elucidated but probably involves hypothesized paratenic hosts such as amphibians, birds, insects, lizards, and rodents, and invertebrate gastropod fauna as obligate intermediate hosts in which the larvae developmental process occurs from first-stage larvae (L1), second-stage larvae (L2), and final infective third-stage larvae (L3) (Figure 1), as is the case for several metastrongyloid parasites in the Angiostrongylidae family.

An extensive survey to identify *G. paralysans* larval stages was conducted in terrestrial gastropods (*n* = 835) collected from a previous feline hotspot located in southern Chile (Valdivia) showed that neither semi-nested PCR, enzymatic digestion nor histopathological examination could identify the presence of *G. paralysans* larvae in mollusks of the families Arionidae, Limacidae, Helicidae, and Milacidae [22]. Therefore, the obligate intermediate hosts of this parasitic disease are still unknown, and the morphological traits of larvae have not been identified so far. Notwithstanding the lack of epidemiological and life cycle knowledge of *G. paralysans,* it has been known since 1930s that the kodkod is the natural definitive host, allowing the pulmonary development of the parasite, while domestic cats are accidental hosts [19,20]. Adult male specimens of *G*. *paralysans* have a total body length of 12–18 mm and a width that varies from 72–103 μm to 26–32 μm anterior to the bursa and the cephalic region, respectively. Consistently larger female adults show a total body length ranging from 27 to 28 mm [21]. The parasite adult stages are located in the definitive host’s meningeal veins and spinal cord subarachnoid space [23], where undeveloped eggs passed by the females, and 16-cell embryos eggs, have been found in the felid bloodstream [16]. The morphometrical measurements of known parasitic stages are summarized in Table 1. To the best of our knowledge, to date, neither eggs nor larvae have been found in any felid host gastrointestinal tract, nor in feces [16].

The migration pathways of *G. paralysans* in vertebrate hosts begin when domestic cats or wild felids acquire the infective third-stage larvae (L3) through ingestion of intermediate/paratenic hosts. Infective larvae penetrate the stomach into the vascular bed, migrating to the portal vein toward the inferior vena cava and/or the thoracic venous system to finally reach the spinal cord via the vertebral venous plexus and the intervertebral veins [30]. Understanding the current lifecycle brings up many questions, and therefore *G*. *paralysans* remains an extremely neglected parasite.

## 2. Worldwide Distribution Range of Gurltiosis

Until recently, gurltiosis was a parasitic disease found only in South America, distributing from Aysén in southern Chile to the northernmost report located in Antioquia, Colombia [31,32]. One confirmed case of gurltiosis has been reported in South America in a domestic cat from the island of Tenerife, and this is also the first report of ophthalmic *G. paralysans* parasitism [33]. However, a necropsy performed at Cornell University on a cat with severe neuropathy revealed an extensive hemorrhagic lesion between the third and sixth lumbar vertebrae, which contained a metastrongyloid female consistent with *G. paralysans* morphologic traits [16]. Those reports possibly constitute imported cases to Spain and the USA, respectively. Feline gurltiosis is often reported as sporadic single case reports and limited sample size studies across South America. The vast majority of reports had a small sample size, (x ≈ 9.4), and were located in rural and suburban areas (Table 2). In South America, the mean age of feline gurltiosis reported cases was 2.5 years old.

A descriptive molecular epidemiology study from a total of 93 domestic cats located in urban areas of Southern Chile, where 54.4% of the studied animals were *G*. *paralysans*-positive, showed the feasible transmission of parasite among domestic cat populations in urban environments and proposed that predictors such as age, sex, lifestyle (indoor/outdoor), anthelmintic use, and cat hunting behavior should be considered as potential risk factors associated with this angio-neurotrophic parasite infection [15]. Thus, those risk factors should be considered in rural, suburban, and urban areas from Argentina, Brazil, Chile, Colombia, Spain, and Uruguay, where *G*. *paralysans*-positive domestic cat cases have been confirmed.

Parasite biological characteristics such as host specificity, life cycle complexity, and climatic tolerance may render parasite species particularly vulnerable or allow them to proliferate [44]. As previously stated, gurltiosis in South America ranges from Colombia to Chile, covering a wide range of environmental conditions. As a result, a climate analysis could provide epidemiological insights into the ecoepidemiology of *G*. *paralysans*. In accordance with the Köppen–Geiger climate classification, the parasite shows great adaptation to different thermal floors and climates because the nematode has been identified in domestic and wild felids located in tropical rainforest (Af), savanna (Aw), hot semi-arid (BSh), humid subtropical (Cfa), oceanic (Cfb), warm-summer Mediterranean (Csb), and tundra (ET) climates [15,32,33,44,45] (Figure 2). Thus, confirmed gurltiosis cases have been reported in tropical, arid (dry), and temperate climates, showing a considerable climatic tolerance. Despite the natural gurltiosis distribution throughout South America, it is essential to note that southern Chile is a well-known gurltiosis hotspot where the main geographic distribution of gurltiosis cases in domestic cats occurs [22]. Meanwhile, the only reports of the disease in wild felids are parasitic myelopathy in a Brazilian margay [21] and the first Wolffhügel report of the disease’s native host, the kodkod [20].

## 3. Clinical Signs and Diagnostic

Feline gurltiosis should always be considered as a differential diagnosis in cats with neurological signs related to thoracolumbar/lumbosacral spinal cord damage [39]. Demonstration and morphological identification of the nematodes in the spinal cord vasculature is the definitive diagnosis. This approach, however, may only be used postmortem [34,37]. Necropsy findings include diffuse sub-meningeal congestion of spinal cord segments (i.e., lumbar, sacral, and coccygeal). The intravascular presence of adult nematodes and larvae stages cause the thickening and congestion of subarachnoid vessels [24,36,43]. Microscopic findings include vascular myelitis and intralesional adult parasites, which principally locate at subarachnoid space in some segments of the spinal cord (third thoracic vertebra to third lumbar vertebrae and fourth lumbar vertebrae to third sacral vertebrae) [38,39]. The gurltiosis *intra vitam* diagnosis is highly challenging due to the lack of evidence in the excretion of *G. paralysans* eggs and larvae stages, neither in feces nor in other body secretions [42], because the environmental stages and parasite transmission route remain unknown. Veterinary clinicians should include exhaustive clinical examination in cats with typical hind limbs neurological signs and complementary tests to approach the diagnosis of this parasitosis. Cats with a clinical history of progressing paraparesis or paraplegia between 2 weeks and 48 months [32,34,37,39], pelvic limb ataxia, tail paralysis, and fecal and urine incontinency in endemic areas should be considered for the diagnosis of feline gurltiosis. 

The neurological signs are related to the neuroanatomical lesions caused by the parasite and include bowel incontinence, urinary dysfunction, tail trembling/atony, pelvic limbs ataxia and tights muscle atrophy and tremor, pelvic limbs tremors, and proprioceptive deficit [37]. The classical chronic myelopathy signs are caused by the vascular proliferation produced by the parasite and result from compression of the white matter in the thoracolumbar and lumbosacral dorsal cord [38]. Some paraclinical findings include non-regenerative anemia, hypochromia, and high blood urea nitrogen (BUN) levels with consequently azotemia [37,46]. Neither eosinophilia nor coagulopathy are common findings in cats with gurltiosis [37]. Ocular lesions and the parasite presence in the fluid-filled space between the cornea and iris (i.e., anterior chamber of the eye) in a domestic cat have also been reported [33]. Paraclinical analysis includes complete blood examination (haemogram), fecal examination, analysis of cerebrospinal fluid (CSF), and imaging (computed tomography, myelography, and magnetic resonance imaging) [37]. Recently, Gomez et al. evaluated the commercial serological Angio Detect™ test (IDEXX™ Laboratories, ME, USA) as a suitable *intra vitam* diagnostic method for feline gurltiosis. They suggested a cross-reaction between *Angiostrongylus vasorum* and *G. paralysans*-specific antigens, which could be used as a new diagnostic tool for feline gurltiosis. Nevertheless, it is necessary to analyze the sensibility and specificity compared with the specific antigen for this parasite to validate the serological test result [42].

Additionally, semi-nested PCR analysis has been proposed as a routinary test for early diagnosis of the parasite in serum. The PCR amplifies a 450 bp fragment of a common metastrongyloid sequence using universal oligonucleotides (please see Table 3). The semi-nested PCR differentiates between *G. paralysans* DNA and *Ae**lurostrongylus abstrusus* with bands of 356 and 300 bp, respectively [43]. It has been proposed that serum samples are more effective than CSF in detecting the parasite by molecular analysis; nevertheless, the result was not statistically significant, and further research is needed [43]. Furthermore, phylogenetic analysis of 28S rRNA (D2-D3 region), ITS1 and ITS2 of the 5.8S rRNA, and partial 18S rRNA sequences demonstrated that *Gurltia* spp. belongs to the family Angiostrongylidae and is therefore morphologically similar to related genera (i.e., *Aelurostrongylus* sp., *Angiostrongylus* sp., *Didelphostrongylus* sp., and *Heterostrongylus* sp.) [21,23]. Additionally, the molecular analysis concluded that *G. paralysans* is most closely related to *A. vasorum* [30]. The parasite D2-D3 region is considered an adequate molecular marker [21,45]. Nevertheless, to date, it has not been possible to identify *G. paralysans*-DNA in fecal samples with any coproparasitological tests, so it is alternatively recommended to use serum or CSF samples [43]. Preliminary analysis also indicates the presence of *G. paralysans* DNA in bronchoalveolar lavage, but it has not yet been validated [15]. Some primers used to determine the presence of *G. paralysans* DNA are listed in Table 2. It is worth mentioning that the accurate *intra vitam* diagnosis of the disease remains highly challenging and more feasible via necropsy in clinically ill felids [18,19,21,30].

## 4. Treatment

Even though the Tropical Council for Companion Animal Parasites (TroCCAP) has affirmed neither a pharmacological treatment schedule nor a therapy that has been proven effective against *G. paralysans* infection in cats [47], some empirical treatments have demonstrated variable degrees of effectiveness against this angio-neurotrophic parasite in felids. Unfortunately, no experimental trial for treating gurltiosis has been conducted to date. However, there are some reports of the successfully antiparasitic drugs used with different effectiveness and clinical improvement results in diagnosed cats (Table 4). For example, the oral (PO) administration of ivermectin resulted in clinical recovery in mild or moderate neurodegenerative feline gurltiosis cases [48]. Moreover, the prophylactic use of other macrocyclic lactones, such as selamectin and milbemycin, may prevent *G. paralysans* infection in cats located in endemic areas [30], together with the control of paratenic/intermediate hosts and responsible pet ownership practices such as keeping the cats indoor and avoiding hunting behavior in gurltiosis hotspot areas could reduce the possibility of acquiring this not yet wholly understand parasitosis [15]. Furthermore, broad-spectrum fenbendazole (benzimidazole) and macrocyclic lactone moxidectin concomitantly administrated with the neonicotinoid imidacloprid may reduce the risk of *G. paralysans* infection, as they do for other related nematode species, such as *A. vasorum* [30].

The PO or subcutaneously (SQ) ivermectin administration in cats is well-tolerated at ranges between 0.2 and 1.3 mg/kg [51,52], the no-effect level is approximately 0.5 to 0.75 mg/kg of body weight, and toxicosis has been reported in a limited number of cats. Routinely, a monthly deworming PO dose of moxidectin of 0.003 mg/kg and a sustained release injectable formulation at a dose of 0.17 mg/kg could be preventively administrated every six months in cats [50]. Additionally, a PO milbemycin monthly dose of 2 mg/kg is available for cats, but adverse effects such as hypersalivation, ataxia, mydriasis, and central nervous system depression should be constantly monitored [53]. Furthermore, the highest safe level of selamectin, which has been proven up to ten times its recommended dose in kittens [49,50], is also a suitable and safe option to treat gurltiosis in domestic cats. However, the PO selamectin administration at the recommended topical dose (6 mg/kg) may cause salivation and vomiting in malnourished or underweight cats [49]. As in other metastrongyloid parasitoses, it is well known that an efficient treatment involves repeated check-ups and repeated treatments when necessary [54]. The use of anti-inflammatory drugs such as non-steroidal anti-inflammatory drugs (NSAIDs) and corticosteroids, neuroprotective vitamins (i.e., Vitamins B, C, E, and K), and other medicaments is left to clinician discretion. The animal’s recovery depends on the gravity of the vascular damage and the grade of limb cord compression. The longer it takes to establish treatment, the lower is the recovery percentage [30].

## 5. The Northernmost *G. paralysans* Case Report in South America

In Colombia, a total of six feline gurltiosis case reports have been documented. Five Siamese cats that lived in the same household (i.e., two kittens, one adult male, and two adult females) from Tarso and one domestic mixed-breed cat from Amagá, Antioquia, exhibited moderate to severe paraplegia with general ataxia, decreasing of superficial sensitivity, and deep sensitivity loss. Additionally, the felids anamnesis evidence progressive paralysis more severe in adults due to the chronic clinical manifestation of the disease. Moreover, the cats showed hind limb atrophy and bladder and bowel dysfunction [32]. All described clinical signs correspond well with the chronic clinical evolution recorded for this angio-neurotrophic parasitic disease [17]. Subsequently, the cats´ necropsy and histopathology analysis showed the presence of *G. paralysans* specimens in the meningeal veins from the tenth thoracic vertebrae to the fourth lumbar vertebrae with medullar compression concomitantly myelomalacia [32]. Locals have frequently reported chronic long-lasting degenerative myelopathy in rural cats in the southwest and eastern Atrato subregion in Antioquia and Chocó, respectively, thereby suggesting *G*. *paralysans* infection in other Colombian areas not yet studied. 

The presence of metastrongyloid parasites such as *Aelurostrongylus abstrusus (Angiostrongylidae)*, *Troglostrongylus brevior (Crenosomatidae)*, *Crenosoma vulpis (Crenosomatidae)*, and the zoonotic *Angiostrongylus vasorum (Angiostrongylidae)* have been reported in gastropods from the *Achatinidae* family in several Colombian regions [55], including Antioquia, where *G. paralysans* cases have been identified [32,55], and the traditional ecological knowledge of peasant communities describing clinical signs in cats compatible with feline gurltiosis (see Figure 3). Additionally, an epidemiological approach showed the patent occurrence of *A. abstrusus* infections in domestic cats from Antioquia [56]. This region has also recently shown the presence of neglected zoonotic parasites such as *Spirometra* sp. *(Diphyllobothriidae)* and *Toxocara cati*
*(Toxocaridae)* in neotropical wild felids [57]. It is essential to highlight that the Gastropoda class has been proposed as an obligate intermediate host for *G. paralysans* and is a highly biodiverse taxon in Colombia with at least 56 families and 120 species spread across various biomes (Appendix A). Terrestrial gastropod species do not only inhabit humid and cool environments; they can also cope and prevail in hot and dry environments, therefore successfully adapting to insolation, heat, and drought [58]. Thus, those findings collectively demonstrate and highlight the epidemiological feasibility of a wider distribution of metastrongyloid nematodes such as *G. paralysans* in Colombia, given that the country has all the climates in which the parasite has been reported. Furthermore, studies on concomitant infections have not been consistently conducted across reported areas, despite their importance in understanding and identifying the epidemiological risk factors associated with feline gurltiosis [15] in domestic cats and wild felid populations.

As mentioned above, the wild felid of the species *Leopardus guigna*, commonly known as kodkod, is the natural definitive host, and domestic cats are considered accidental hosts for gurltiosis [19,20]. The parasitic myelopathy due to *G*. *paralysans* adult nematodes and eggs inside the veins of subarachnoid space in the spinal cord of a margay has been described in Santa Catarina, Brazil [20]. Both the Goeffroy’s cat (*Leopardus geoffroyi*) and the northern tiger cat (*Leopardus tigrinus*) have been proposed as definitive hosts [15,21]. Neither the kodkod nor Goeffroy’s cat are distributed within Colombia; therefore, the margay and the northern tiger cat should be considered potential gurltiosis hosts in this geographical location. Other neotropical wild felid species found in Colombia such as jaguarundi (*Herpailurus yagouaroundi*), ocelot (*Leopardus pardalis*), jaguar (*Panthera onca*), and puma (*Puma concolor*) should not be ruled out as the natural definitive host in the Northernmost South American distribution area, where the parasite has so far been documented (Table 5).

## 6. Conclusions

In parallel with other neglected felid diseases frequently underestimated by veterinary clinicians, gurltiosis should be included in the differential diagnoses of feline spinal cord disorders. The prompt and accurate diagnosis of *G*. *paralysans* will contribute to improved health of infected definitive hosts and result in proper anthelminthic treatments impeding further gurltiosis spreading among felid populations. The poor understanding of this neglected angio-neurotrophic parasite’s life cycle demands further research to identify the potential gastropod intermediate host species and paratenic hosts such as amphibians, birds, insects, lizards, and rodents. Additionally, comprehensive sampling efforts should be developed throughout the Americas, where the majority of feline gurltiosis cases are reported. Particular attention should be taken in North America, Africa, and Europe because cases outside of South America were recently reported in the west Africa palearctic realm and the north Nearctic realm, on the island of Tenerife, Spain, and NY, USA, respectively [16,33]. In conclusion, further studies are needed to evaluate the parasite occurrence among domestic cats and neotropical wild felid species distributed within Colombia as well as the plethora of gastropod/paratenic fauna that may harbor the developing larvae (L1–L3) stages of this underestimated parasite.

## Figures and Tables

**Figure 1 pathogens-10-01601-f001:**
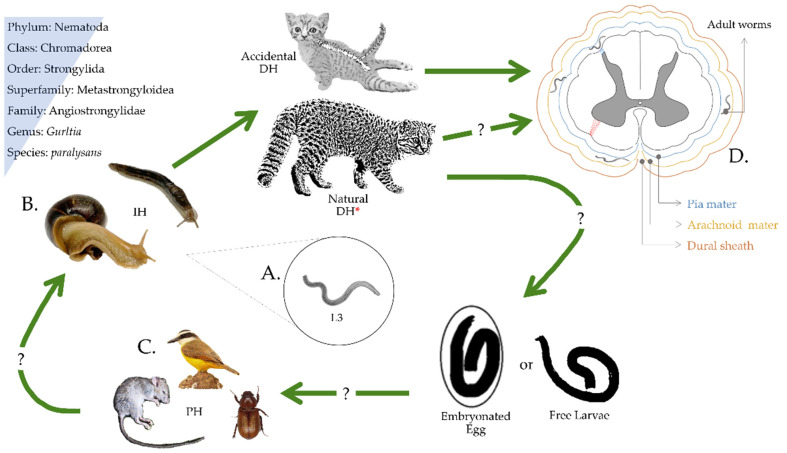
Proposed life cycle of *Gurltia paralysans*. Felids become infected by consuming the L3 larvae (**A**) from an infected intermediate host (**B**) or paratenic host (**C**). L3 migrates from the intestinal tract to the central nervous system and invades the veins of the subarachnoid space (**D**) of the spinal cord. There, larvae mature to adult worms and produce eggs and reproduce through eggs. The elimination route of the eggs or first larval stage (L1) or how the intermediate host becomes infected with L1 is still unknown. * Definitive host.

**Figure 2 pathogens-10-01601-f002:**
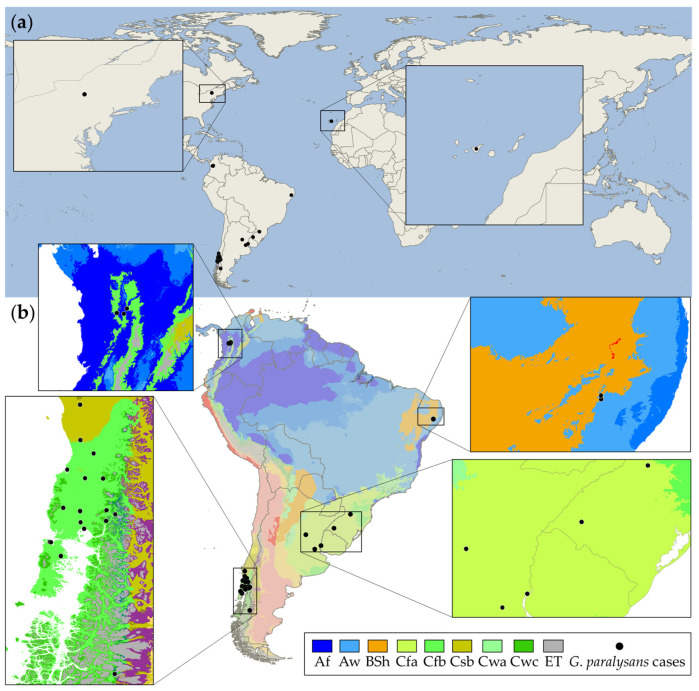
Worldwide reported cases of feline gurltiosis in domestic cats and wild felids. (**a**) Enlargement of the areas outside South America (i.e., Tenerife Island, Spain, and New York, NY, USA) where imported cases have been recorded. (**b**) Köppen–Geiger climate classification from South American case reports of the angio-neurotrophic parasite *G. paralysans*. Close-up location of the parasite reports from Colombia, Brazil, and the triple border area between Brazil, Uruguay, and Argentina. Additionally, feline gurltiosis hotspot area in Southern Chile where the most significant number of occurrence disease cases have been reported. Af: Tropical rainforest, Aw: Savanna, Bsh: Hot semi-arid, Cfa: Humid subtropical, Cfb: Oceanic, Csb: Warm-summer Mediterranean, Cwc: Monsoon-influenced subpolar oceanic, Cwa: Monsoon-influenced humid subtropical, ET: Tundra.

**Figure 3 pathogens-10-01601-f003:**
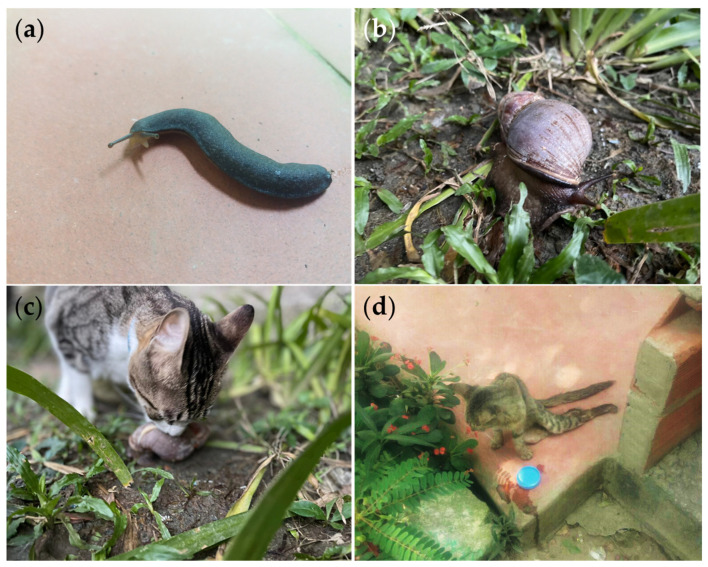
Ecoepidemiological factors that allow gurltiosis occurrence in the Colombian Andean region. Images of *G. paralysans* gastropod intermediate hosts (IH) such as (**a**) smooth land slugs of the genus *Deroceras* sp. found in the Southwest subregion of Antioquia, Colombia and (**b**) Giant African snail (*Lissachatina fulica*) in an urban environment (Medellín city). (**c**) The natural domestic cat’s curious behavior on a Giant African land snail favors close interaction with IH, demonstrating the feasibility of parasite transmission in urban areas. (**d**) Clinically ill mixed-breed domestic cat in an Andean rural area with long-lasting degenerative myelopathy and severe paraplegia.

**Table 1 pathogens-10-01601-t001:** Comparative morphometrical traits of *Gurltia paralysans* adult stages among related Angiostrongylidae nematodes.

	*G. paralysans* ^a^	*Ael. Abstrusus* ^b^	*Ang. Vasorum* ^c^
**Male**			
Body length	12,000–18,000 μm	5440–7080 μm	14,000–16,000 μm
Body width	72–103 μm	42.5–92.7 μm	–
Esophagus length	360–432 μm	219.4–360.9 μm	219 μm
Spicule length	650–816 μm	103.7–138.9 μm	360–490 μm
Gubernaculum length	37–39 μm	18.8–31 μm	34–45 μm
**Female**			
Body length	20,500–30,000 μm	7950–10,587 μm	15,000–21,000 μm
Vulva to tail tip	102–150 μm	223.6 μm	–
Tail length	30–50 μm	27–29 µm	27–29 µm
Eggs	40–72 × 26–54 μm	37.8–48.4 × 94.5–99.6 μm	70–80 × 40–50 μm

References ^a^ [21,22,23], ^b^ [24,25], ^c^ [25,26,27,28,29].

**Table 2 pathogens-10-01601-t002:** Chronologic confirmed case report of the angio-neurotrophic *Gurltia paralysans* parasite.

Year	Location	Area	Host	(*n*)	Diagnosis Method	Reference
1933	Chile	Rural	*Felis catus*	-	M	[18]
1933	Chile	Rural	*Leopardus guigna*	-	-	[19]
2010	Chile	Rural	*Felis catus*	4	Ct, Hp, M	[34]
2011	Uruguay	Rural	*Felis catus*	2	Hp, M	[35]
2011	Colombia	Rural	*Felis catus*	6	Ct, Hp	[32]
2011	Argentina	Suburban	*Felis catus*	1	Hp	[36]
2012	Chile	Rural	*Felis catus*	3	Hp, M	[23]
2013	Chile	Rural	*Felis catus*	9	Ct	[37]
2013	Brazil	Rural	*Felis catus*	4	Hp	[38]
2016	Chile	Rural	*Felis catus*	1	Hp	[39]
2016	Argentina	Rural	*Felis catus*	3	Hp, M	[40]
2017	Chile	Suburban/Rural	*Felis catus*	-	SEM, phylogeny	[21]
2018	Spain	Suburban	*Felis catus*	1	M, phylogeny	[33]
2019	Brazil	Rural	*Felis catus*	7	Hp	[41]
2020	Chile	Rural	*Felis catus*	4	Serology, M	[42]
2020	Chile	Suburban	*Felis catus*	1	PCR	[31]
2020	Chile	Rural	*Felis catus*	7	PCR	[43]
2020	Brazil	Rural	*Leopardus wiedii*	1	M	[20]
2021	Chile	Urban	*Felis catus*	93	PCR	[15]

M: morphology, Ct: Computed tomographic-myelography, Hp: histopathology, SEM: Scanning electron microscopy. PCR: Polymerase chain reaction.

**Table 3 pathogens-10-01601-t003:** Reported PCR analysis for *Gurltia paralysans* proposed by López-Contreras et al., 2020, and Barrios et al., 2021.

PCR Type & Gene	Primer Set (5′→3′)	Amplicon Size (bp)	Identification
semi-nested PCR 28S rDNA	AaGp28Sa1-R: AGGCATAGTTCACCATCT AaGp28Ss1-F: CGATRATATGTATGCCATT	450	Common sequence Metastrongyloidea
	E1:Aa28Ss2-F: CGTTGATGTTGATGAGTATC E2:Gp28Sa3-R: TCTTGCCGCCATTATAGTA	300 356	*A. abstrusus* *G. paralysans*
Endpoint-PCR rDNA Feline	F: AGCAGGAGGTGTTGGAAGAG R: AGGGAGAGAGCCTAATTCAAAGG	100	Internal control

**Table 4 pathogens-10-01601-t004:** Suitable pharmacological drugs for treating the metastrongyloid *Gurltia paralysans* in domestic cats.

Drug	Structure ^§^	Dose ^†^	Route and Frequency of Administration	Reference
Ivermectin ^a,^*	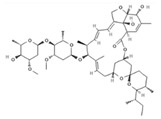	0.2–0.4	PO, SID, 1× weekly, 4 doses	[30,32,48]
Selamectin ^b^	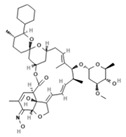	6	PO, SID	[30,49]
Milbemycin oxime ^c^	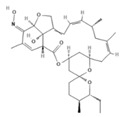	2	PO, SID, 1× monthly	[30,50]
Ricobendazole ^d,^*	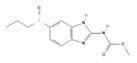	20	PO, SID, 2× days	[30]
Fenbendazole ^e^	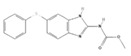	50	PO, SID, 1× day, 3–5 doses	[30]

^†^ The doses are indicated in mg (milligrams) per Kg (kilogram) of body weight. ^§^ Chemical structure depiction adapted from PubChem^™^. ^a^ 22,23-Dihydroavermectin B1a. ^b^ 25-cyclohexyl-25-de(1-methylpropyl)-5-deoxy-22 23-dihydro-5-(hydroxyimino)-avermectin B1 monosaccharide. ^c^ Milbemycin a4 5-oxime. ^d^ Methyl [5-(propane-1-sulfinyl)-1H-benzoimidazol-2-yl]-carbamate. ^e^ 2-(Methoxycarbonylamino)-5-(phenylthio)benzimidazole. * Drugs successfully administered to *G. paralysans* diagnosed cats.

**Table 5 pathogens-10-01601-t005:** Potential *Gurltia paralysans* definitive wild felid host species in South America.

Genus	Species	Common Name	Conservation Status ^§^
*Herpailurus*	*yagouaroundi* ^1^	Jaguarundi	LC
*Leopardus*	*colocolo*	Pampas cat	NT
*Leopardus*	*geoffroyi* ^2^	Geoffroy’s cat	LC
*Leopardus*	*guigna* ^2^	Kodkod	VU
*Leopardus*	*guttulus*	Southern tiger cat	VU
*Leopardus*	*jacobita*	Andean cat	EN
*Leopardus*	*pardalis* ^1^	Ocelot	LC
*Leopardus*	*wiedii* ^1,2^	Margay	NT
*Leopardus*	*tigrinus* ^1^	Northern tiger cat	VU
*Panthera*	*onca* ^1^	Jaguar	NT
*Puma*	*concolor* ^1^	Puma	LC

^1^ Felid species distributed within Colombia. ^2^ Species in which the parasite has been reported. ^§^ Based on the IUCN threat levels of classifications for endangered species. LC: Least-concern, NT: Near threatened, VU: Vulnerable, EN: Endangered.

## Data Availability

Not applicable.

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
