# Peer review of "The Neglected Angio-Neurotrophic Parasite Gurltia paralysans (Nematoda: Angiostrongylidae): Northernmost South American Distribution, Current Knowledge, and Future Perspectives"

_pathogens, 2021, doi:10.3390/pathogens10121601_

Round 1

Reviewer 1 Report

Overall, nice summary of this parasite. I admit I had not heard of it previously, and it was nice to be educated on it. I will look forward to the full special edition to learn even more. A few comments below.

Line 77: Think you are missing a word or words in this sentence ‘and remains G. paralysans’ doesn’t make sense

Lifecycle –  line 56 you state rodents, and other animals, as paratenic hosts where larval development occurs. But a paratenic host is which no development occurs, they are like carrier species. A larvae may migrate and become encysted in these hosts to await ingestion by a definitive host. Generally, if larval development is occurring in these hosts they would be intermediate hosts, not paratenic.

Have any larval stages been pictured? Have rodents actually been identified as infected or just assumed?

If the kodkod is the definitive host, have eggs and/or larvae been identified in kodkod faeces? Or on what basis have they been so defined?

I’d like to see a phylogenetic tree with whatever sequenced examples of G. paralysans exist in relation to some of the other closely related species (or presumed closely related species).

Line 124: Would these temp ranges be inhibitory of some mollusk species? The giant African land snail has successfully colonized in many locations across south America, has it been investigated as a potential host? Has G. paralysans been identified in Africa? Has anyone looked for it?

Line 152: unknown

Line 180: Proposed by whom and on what basis?

Line 190: This is the authors own analysis/research? Perhaps state that preliminary analysis by the authors also indicates etc etc and then put in brackets the author initials for who is doing this work.

Line 286-288: This should be introduced in the lifecycle section.

Author Response

Dear Reviewer;

We would like to thank you for highlighting the significant contribution of the work for a better comprehension of this extremely neglected felid parasite. The Line 77 sentence was corrected. Regarding the intermediate and paratenic host roles in proposed lifecycle and the larvae development correspond amendment was made as equal as proper rewording of the sentence in agreement with your comment. Both the paratenic (amphibians, birds, insects, lizards, and rodents) and intermediate gastropod hosts are hypostatized, since to date first-stage larvae (L1), second-stage larvae (L2) and final infective third-stage larvae (L3) have not been identified naturally infecting any kind of host, therefore the morphological traits of these stages is unknown. However, based on your recommendation, a comparative morphology table of Gurltia paralysans adult stages among related Angiostrongylidae nematodes was included. Additionally, no eggs or larvae have been found in felid feces. However, both adults and eggs in different stages of development were found inside the lumbar spinal cord vein lumen of wild/domestic felids.

Other authors have previously published phylogenetic analysis of G. paralysans based on MAFFT alignments for the 28S D2–D3 and ITS2 rDNA region where other metastrongyloid genera were included (10.1016/j.vprsr.2017.10.001, http://dx.doi.org/10.4067/S0719-81322021000100033). We hope to soon be able to contribute with phylogenetic analysis with own genetic sequences of the parasite obtained from Colombian felines.

The world “unknow” in Line 152 was changed to “unknown” in the same way the rewording of the concept and proper citation of Line 180 and Line 190 sentences were made. Finally, Lines 286-299 were moved to section “1. Introduction, Brief History, and the Enigmatic Life Cycle” and added to lifecycle paragraph as you indicated us. Regarding Line 124 quotation, temperature does not affect different organisms equally (10.1242/jeb.037473) and does not affect the same organism equally at all stages of its life cycle (10.1126/science.1163156), but the vast majority of land snails have developed strategies to successfully cope with hot and dry climate (10.1002/ece3.5607). Unfortunately, in Colombia the giant African snail (Lissachatina fulica) has expanded across different altitudinal ranges and climates. An extensive survey to identify G. paralysans larval stages was conducted in terrestrial gastropods from families Arionidae, Limacidae, Helicidae, and Milacidae in Chile (10.1590/S1984-296120201087). Was identified as natural intermediate host of Aelurostrongylus abstrusus, Angiostrongylus vasorum, Troglostrongylus brevior, and Crenosoma vulpis in Colombia but to date G. paralysans has not been identified in any snail or slug (10.1371/journal.pntd.0007277), thus the intermediate/paratenic host of the parasite still remaining unknown.

Best regards.

Reviewer 2 Report

This is an interesting MS highlighting a neglected parasite with excellent detail on what is currently known and the gaps in our knowledge.

The MS requires careful attention to the written English.

My suggestions are minor in nature.

Line 14-15: Rephrase the sentence beginning “The current life cycle…

Line 16: explain “intra vitam

Line 18: unknown

Line 20: clinical improvement

Line 22: in addition to the gastropod fauna

Line 24: order key words more logically – start with the species name, then the disease, then the hosts etc

Line 49: Perhaps refer to the very useful Table 4 here when Kodkod is first mentioned

Line 57: An extensive survey to identify the larval stage of the parasite was conducted in….

Line 58: semi-nested and check consistency throughout the MS

Line 61: this parasite not the disease

Line 66-67: clarify re the egg stages here

Line 68: no eggs or larvae

Line 69: explain “intraorganic”

Line 69-71: Evidence from a laboratory model?

Figure 1: Larvae mature to adult worms and produce eggs (legend); arrows require better definition

Line 89: explain L3 and L6 (and T10 to L4 in line 264)

Table 1: explain – (missing data?)

Line 111: Koppen:Geiger classification – explain a little more

Line 133: Clinic (title) – Clinical signs?

Line 176: What result? Ambigious sentence

Table 3: What is Posology?

Line 235: 5 cats that lived in the same household

Line 242: Long-lasting evolution? Unclear

Table 4: Explain abbreviations in the column named conservation status

Author Response

Dear Reviewer;

First, we would like to thank you for highlighting the detail level of current review manuscript and the effort to contribute to existing conceptual gaps of the parasite and propose research perspectives about this neglected parasite, and the usefulness of presented information. Sentence in Line 14-15 was rephrased. Regarding the “intra vitam” concept first used in manuscript Line 16 is a synonym of intravital and common adjective used to describe an event that occurred during life. In contest “intra vitam” along the manuscript describes different diagnostic techniques of gurltiosis identification in living cats as ante-mortem paraclinical examination test. This concept has been used not only in feline gurltiosis but also in other infectious deiseses (10.3390/pathogens9110921, 10.3389/fvets.2020.591444, 10.1017/s0031182003003500). The sign “-“ used in Table 1 and Table 2 indicated absence of the data and unreported information in cited references. Additionally, Lines 18, 20, 22, 24, and 176 were reworded and corrected in agreement to your comments. Key words in Line 24 were organized was you suggest. The phrase in Line 57 was changed to “An extensive survey to identify the larval stage of the parasite was conducted in….”. To avoid confusion along the manuscript, the term “intraorganic” was eliminated. Lumbar/thoracic vertebrae nomenclature was modified, and abbreviation “L” was only used to refer to the larvae stages of the parasite. The use of “semi-nested” was properly checked and consistent throughout the text. The section three title was changed to “3. Clinical signs & Diagnostic”. The term “posology” in Table 3 was changed to “Route and frequency of administration”, and Line 235 phrase change to “…five cats that lived in the same household…”. The unclear “Long-lasting evolution” concept in Line 242 change to “…chronic clinical evolution…”, and conservation status abbreviations in Table 4 were duly explained.

Warmest regards.

Reviewer 3 Report

Dear authors,

Thank you for this good review about a neglegded helminth parasite of cats.

If you could, it will very valuable to add a section on this parasite different stages morphology in details and illustrations, with special focus on its unique characterstic features.

Also, a section on its taxonomic status and relation to other metastrongyloids in different animals.

Thanks.

Regards.

Author Response

Dear Reviewer;

First, we would like to thank you for such a precise and clear comments to improve the manuscript, and for recognize the effort and usefulness of presented information about this extremely neglected felid parasite. Based on your indications the revised version of the manuscript includes the Table 1 in section “1. Introduction, Brief History, and the Enigmatic Life Cycle”, which describe morphological traits of Gurltia paralysans adult stages among related Angiostrongylidae nematodes. Since to date first-stage larvae (L1), second-stage larvae (L2) and final infective third-stage larvae (L3) have not been identified naturally infecting any kind of host, therefore the morphological traits of these stages are still unknown. Other authors have previously published phylogenetic analysis of G. paralysans based on MAFFT alignments for the 28S D2–D3 and ITS2 rDNA region where other metastrongyloid genera were included (10.1016/j.vprsr.2017.10.001, http://dx.doi.org/10.4067/S0719-81322021000100033). We hope to soon be able to contribute with phylogenetic analysis with own genetic sequences of the parasite obtained from Colombian felines.

With kind regards.